# Sensor-Based Assessment of Mental Fatigue Effects on Postural Stability and Multi-Sensory Integration

**DOI:** 10.3390/s25051470

**Published:** 2025-02-27

**Authors:** Yao Sun, Yingjie Sun, Jia Zhang, Feng Ran

**Affiliations:** 1School of Physical Education, China University of Mining and Technology, Xuzhou 221116, China; sunyao999999@126.com (Y.S.); syj5199@163.com (Y.S.); 2School of Physical Education, Chongqing University, Chongqing 401331, China; zhangjiaaa@cqu.edu.cn

**Keywords:** mental fatigue, postural control, sensory reweighting, center of pressure, discrete wavelet transform, sample entropy

## Abstract

Objective: Mental fatigue (MF) induced by prolonged cognitive tasks poses significant risks to postural stability, yet its effects on multi-sensory integration remain poorly understood. Method: This study investigated how MF alters sensory reweighting and postural control in 27 healthy young males. A 45 min incongruent Stroop task was employed to induce MF, validated via subjective Visual Analog Scale (VAS) scores and psychomotor vigilance tests. Postural stability was assessed under four sensory perturbation conditions (O-H: no interference; C-H: visual occlusion; O-S: proprioceptive perturbation; C-S: combined perturbations) using a Kistler force platform. Center of pressure (COP) signals were analyzed through time-domain metrics, sample entropy (SampEn), and Discrete Wavelet Transform (DWT) to quantify energy distributions across sensory-related frequency bands (visual: 0–0.1 Hz; vestibular: 0.1–0.39 Hz; cerebellar: 0.39–1.56 Hz; proprioceptive: 1.56–6.25 Hz). Results: MF significantly reduced proprioceptive energy contributions (*p* < 0.05) while increasing vestibular reliance under O-S conditions (*p* < 0.05). Time-domain metrics showed no significant changes in COP velocity or displacement, but SampEn decreased under closed-eye conditions (*p* < 0.001), indicating reduced postural adaptability. DWT analysis highlighted MF’s interaction with visual occlusion, altering cerebellar and proprioceptive energy dynamics (*p* < 0.01). Conclusion: These findings demonstrate that MF disrupts proprioceptive integration, prompting compensatory shifts toward vestibular and cerebellar inputs. The integration of nonlinear entropy and frequency-domain analyses advances methodological frameworks for fatigue research, offering insights into real-time sensor-based fatigue monitoring and balance rehabilitation strategies. This study underscores the hierarchical interplay of sensory systems under cognitive load and provides empirical evidence for optimizing interventions in high-risk occupational and clinical settings.

## 1. Introduction

Postural control (Please refer to the Appendix A for the explanation of the full text terms.) is a basic element in human motor behavior. It combines multiple sensory inputs, such as visual, vestibular, and proprioceptive inputs, to ensure stability [1]. Despite the importance of postural control in daily life, especially in situations that require long-term mental engagement (e.g., driving, operating machinery, or high-precision tasks), the effect of mental fatigue on postural control has not been adequately investigated. Mental fatigue is a psychological state caused by prolonged cognitive activity. It is known to negatively affect motor and cognitive performance, but its specific influences on the integration and prioritization of sensory information during postural control remain unclear [2]. Sports physiology defines this phenomenon as cognitive performance fatigue or state fatigue when referring to both subjective and objective perceptions [3]. In recent years, numerous studies have attempted to clarify the specific effects of mental fatigue. For instance, many studies have confirmed that mental fatigue has a negative effect on endurance sports performance [4,5,6]. Martin et al. further explored the underlying mechanism and proposed that the effect is due to an individual’s increased perception of effort, which involves changes in the activities of multiple brain regions [7], and may be affected by environmental factors and the degree of fatigue [8]. At the same time, some research suggests that, unlike the significant decrease in endurance performance, mental fatigue does not affect individual performance in high-intensity anaerobic exercise, including whole-body endurance, muscle endurance, maximal strength, and power [9,10].

Prior studies have investigated the effects of fatigue on postural stability, primarily focusing on physical fatigue. For example, Gebel et al. demonstrated that physical fatigue can significantly disrupt time-domain postural sway indicators [11], whereas Morris et al. found gender- and age-related differences in postural control before and after fatigue. These studies provide important insights but lack a detailed examination of mental fatigue’s unique mechanisms and its impact on sensory compensation [12]. Specifically, studies on mental fatigue have produced inconsistent findings regarding its influence on postural control. While Hachard et al. observed a significant decrease in sample entropy under mental fatigue conditions, other studies reported non-significant changes in key sway metrics [13]. These discrepancies may stem from differences in experimental design, participant demographics, and methods of fatigue induction. Furthermore, traditional sway metrics, such as center of pressure (COP) velocity or displacement, are often employed to evaluate postural stability [14]. Although these metrics are widely used, they may fail to capture subtle changes in postural control dynamics resulting from mental fatigue. Nonlinear metrics, such as sample entropy, were recently proposed to better reflect the complexity and variability of postural control signals [15]. However, few studies have investigated how mental fatigue affects the frequency-domain energy proportions of sensory signals, particularly those corresponding to visual, vestibular, cerebellar, and proprioceptive systems under multi-sensory interference conditions. Given these gaps, this study aims to address the following questions: How does mental fatigue affect postural stability across various sensory interference conditions? What are the compensatory mechanisms employed among sensory systems under mental fatigue? By analyzing COP dynamics through time-domain and frequency-domain indicators, this study provides novel insights into the compensatory adjustments in sensory input during mental fatigue. This research not only addresses critical limitations in the existing literature but also contributes to the development of sensor-based systems for monitoring fatigue and its effects in applied environments. The findings have implications for both theoretical models of multi-sensory integration and practical applications, such as occupational ergonomics and rehabilitation therapies.

Therefore, further exploration is necessary to uncover how mental fatigue influences postural stability under multi-sensory interference conditions, highlighting the need for advanced signal processing techniques to extract detailed insights from COP data. This study employs DWT as a primary analytical tool to investigate postural control under mental fatigue. DWT enables multiscale analysis of signals and is widely recognized for its ability to decompose complex, non-stationary time-series data into distinct frequency bands [16]. Compared to traditional Fourier Transform, DWT not only captures global features but also preserves localized temporal changes, making it particularly effective for detecting dynamic shifts in postural control caused by mental fatigue. Specifically, this approach is well suited for analyzing COP signals, as it allows for the isolation of energy changes across sensory-related frequency bands, namely the following: ultra-low frequencies (0–0.1 Hz) for visual input, very low frequencies (0.1–0.39 Hz) for vestibular input, low frequencies (0.39–1.56 Hz) for cerebellar input, and moderate frequencies (1.56–6.25 Hz) for proprioceptive input. By decomposing COP signals into these physiological frequency bands, DWT provides a nuanced perspective on how mental fatigue alters the energy proportions of sensory inputs during postural control tasks. This capability addresses the limitations of conventional time-domain or static metrics that may overlook critical features of multi-sensory compensation and integration under fatigue conditions. At the same time, to systematically evaluate the effects of mental fatigue on postural control, this study introduces three key independent variables [17], open-eye/closed-eye conditions, proprioceptive perturbations, and combined sensory disruptions, each selected based on their theoretical and practical relevance. Based on this, this study employs cognitive interventions to induce mental fatigue. The DWT method is utilized to quantitatively compare the spectral energy proportions of different frequency bands corresponding to various sensory inputs under each condition. This research serves as a foundational exploration of the mechanisms through which fatigue affects posture control, with the goal of uncovering some of the ways in which mental fatigue influences human posture control. It provides a reference for subsequent in-depth studies on posture control.

## 2. Objective and Methods

### 2.1. Participants

This study recruited 27 male university students, aged 18–25, on a random basis for participation in the experiment. The screening criteria required that the subjects had no prior systematic competitive sports training or experience as professional athletes. They also could not have any symptoms of lower limb injuries within the past three months, degenerative joint diseases, visual and vestibular sensory disorders, sleep disorders, neurological diseases, or psychiatric disorders. It was necessary that participants were in good health, with normal joint mobility in their lower limbs and normal exercise capacity. Additionally, to ensure that significant sleep disturbances did not affect the experimental outcomes, participants needed to pass the Pittsburgh Sleep Quality Test screening, with a score of less than 7 (PQSI < 7), thereby ensuring the validity of the associated cognitive experiments. All participants signed informed consent forms before the trial. The study was conducted in accordance with the Declaration of Helsinki, and approved by the Institutional Review Board of School of Physical Education, China University of Mining and Technology(protocol code 2024029 and date of approval 5 January 2024).

The sample size was calculated using G*power software 3.1.9.2 [18]. The effect size was slightly larger than a medium effect size, based on preliminary experiments and related prior literature. For this experiment, the effect size was set at 0.26, with an α error value of 0.05. A sample size test was conducted for a single-group three-factor repeated measures ANOVA. Under these conditions, the recommended sample size is 26 people. In total, 27 participants were included. The basic information of the participants is shown in Table 1.

### 2.2. Experimental Method

#### 2.2.1. Instrumentation and Equipment

Three-dimensional force platform [19]: This study employed two Swiss-made Kistler three-dimensional force platforms(KISTLER Switzerland, Winterthur, Switzerland) (90 cm × 60 cm × 10 cm × 2) (Model: 9287B) to measure forces along the *x*, *y*, and *z* axes. The platforms operated at a sampling frequency of 1000 Hz. Data collection was streamlined using the Vicon Nexus 1.7 software, which calculated pertinent mechanical parameters.

E-prime 3.0 software was utilized to develop a four-color, 100% inconsistent Stroop program, which is paired with a correspondingly colored marker key keyboard [20].

Additional apparatus: An electronic height gauge, a weighing scale, a tape measure, a sponge pad, and a gaze board stand were used. See Figure 1.

#### 2.2.2. Experimental Procedure and Experimental Design

(1) Experimental Prerequisites: Prior to the commencement of the experiment, participants were mandated to adhere to the following prerequisites: (1) ensure a minimum of 7 h of sleep within the 24 h preceding the experiment; (2) abstain from engaging in vigorous physical activity and consuming alcohol within the 24 h leading up to the experiment; and (3) avoid the intake of caffeine and nicotine within an 8 h window before the experiment. Additionally, participants were required to complete the Pittsburgh Sleep Quality Index questionnaire survey between 12 and 24 h prior to the official start of the experiment. Upon arrival at the experimental venue, it was imperative that participants completed the pre-experiment questionnaire and executed the informed consent form before the initiation of the experiment.

The experiment was conducted under strictly controlled environmental conditions, with consistent lighting arrangements, a stable indoor temperature of 24 °C, and no noise disturbances. All experimental procedures were carried out between 14:00 and 20:00 to mitigate the effects of time [21]. The guidance task configurations remained identical across all sessions. The sequence of sensory condition interference varied randomly among individuals, as determined by a random number table. However, this order was maintained consistently before and after the mental fatigue intervention.

(2) Testing Protocol for Upright Posture Control Under Visual and Proprioceptive Perturbation Conditions.

The experimental procedure for the upright posture control test, conducted under conditions of visual and proprioceptive interference, involved a static bipedal stance test. The subject, following a demonstration by the experimenter, stood with feet apart on the force platform’s center, undergoing four randomized sensory condition interference tests. These tests utilized the open-eyes/closed-eyes method for visual occlusion and a 40DD medium density sponge pad to induce a certain degree of proprioceptive interference [22]. The four conditions were as follows: open-eyes with a hard force platform surface (O-H), open-eyes with a soft surface (O-S), closed-eyes with a hard force platform surface (C-H), and closed-eyes with a soft surface (C-S). When their eyes were open, the subjects were instructed to fixate on a point set 3 m ahead.

Experimental Instructions: The instructions provided to the subjects during upright stance control tasks significantly influenced the outcomes. Depending on these instructions, most of the studied COP parameters exhibited changes ranging from 8% to 71% [23]. It was observed that the highest consistency in results was achieved when the subjects were instructed to “stand as still as possible”. Consequently, this instruction was the sole requirement for subjects during the testing phase. The duration of each experiment was set at 70 s per trial, with a total of three valid measurements for each item. This resulted in an overall experimental time of approximately 20 min.

(3) Protocol for Mental Fatigue Induction.

Based on prior studies, the Stroop experiment had proven to be highly reliable in inducing a state of mental fatigue [24]. To minimize the effects of learning or increased compensatory cognitive effort, the current study employed a 45 min four-color Stroop task with 100% incongruent cognitive intervention. This meant that all trials in the task were incongruent, with the font colors of the color words not matching their semantic meanings. The participants were required to ignore the semantic meaning and respond based solely on the font color. Specifically, the experiment involved four color words, yellow, red, green, and blue, presented in incongruent font colors as follows:-The word “yellow” was displayed in red, green, or blue font.-The word “red” was displayed in yellow, green, or blue font.-The word “green” was displayed in yellow, red, or blue font.-The word “blue” was displayed in yellow, red, or green font.

Experimental Procedure: Initially, during the preparation phase, the task requirements were explained to the participants to ensure their understanding. A brief practice session was conducted to familiarize them with the task. In the formal experimental phase, the participants were seated in front of a computer with their heads positioned 80–100 cm from the monitor. They placed their non-dominant hand on the keyboard and initiated the task by pressing the spacebar upon hearing the command “start.” The participants were required to press the key corresponding to the font color as quickly as possible (e.g., the yellow key for yellow font, the red key for red font). Each test set consisted of 100 trials lasting a total of 300 s. After completing one set, the participants took a 20 s break before immediately starting the next set, for a total of eight sets. The entire fatigue intervention lasted approximately 45 min. Upon completion of all eight sets, a state of mental fatigue was considered to be achieved. If the continuity of the fatigue intervention was interrupted for any reason, the experiment was terminated. The detailed experimental procedure is illustrated in Figure 2. Mental fatigue tests were conducted immediately after the fatigue intervention and again 20 min later.

(4) Evaluation Index.

1. The time-domain-specific indicators included the following: COP anterior–posterior average velocity (Velocity_AP), movement distance (Length_AP), and sway index (DI_AP); COP medial–lateral average velocity (Velocity_ML), movement distance (Length_ML), and sway index (DI_ML). Additionally, the overall indicators encompassed COP total average velocity (Velocity), total sway area (Area), and trajectory length per unit area (Length_per_area).

2. Frequency-Domain Energy: The energy proportions of the four frequency bands, namely visual (0–0.1 Hz, ultra-low), vestibular (0.1–0.39 Hz, very low), cerebellar (0.39–1.56 Hz, low), and proprioceptive (1.56–6.25 Hz, moderate), were represented as E_Vision, E_Vestibule, E_Cerebellum, and E_Proprioception, respectively.

3. Additional Auxiliary Indices: The sample entropy of the COP in both AP and medial–lateral directions were denoted as SampEn_AP and SampEn_ML, respectively.

4. Subjective and Objective Indicators of Mental Fatigue: (1) Visual Analog Scale (VAS): A specifically designed VAS was utilized for this study. For subjective assessments of mental fatigue, the participants were informed that the two ends of the cursor represented a spectrum from “very relaxed, no mental fatigue at all” to “mentally exhausted and unable to persist”. When assessing motivation, the cursor’s two ends represented “full of energy” to “loss of motivation, giving up the experiment”. For effort perception, the ends of the cursor denoted “I have hardly made any effort” to “I have given my best effort”. Subjective reports were gathered before and after the upright posture control task, both before and after fatigue intervention. The participants manually adjusted the cursor on a 10 cm horizontal ruler to indicate their subjective feelings at the time of testing. To prevent bias, the specific scale was concealed and only visible to the tester. After data collection, it was categorized into subjective evaluation indicators for mental fatigue, motivation, and effort perception. (2) The psychomotor vigilance task (PVT), specifically the fastest 10% reaction time (PVT_10%RT), was administered at the conclusion of each subjective reporting phase. This involved the use of the E-prime 3 min PVT program to gather reaction time data for the corresponding period. Following data processing, the fastest 10% reaction time (PVT_10%RT) was derived and utilized as an objective measure for assessing mental fatigue.

(5) Comprehensive Experimental Methodology:

The comprehensive procedure, utilizing the aforementioned experimental scheme, is depicted in Figure 3. All experiments meticulously regulated the indoor environmental factors such as temperature, lighting, and sound, with a cumulative duration of approximately 1 h and 50 min. The margin of error in terms of time for each experiment did not exceed 5 min.

### 2.3. Mathematical Statistics

Data were analyzed using MATLAB 2023 (MathWorks) to extract COP metrics, followed by statistical processing in SPSS 26.0. The results are expressed as mean ± SD. Paired *t*-tests assessed mental fatigue scores and PVT reaction times. Parametric/non-parametric comparisons of pre–post-fatigue time-domain indicators used paired *t*-tests and Wilcoxon tests, respectively. One-way and three-way repeated measures ANOVA evaluated sensory interference effects and multi-frequency energy interactions (mental fatigue × visual occlusion × proprioceptive interference). Primary outcomes focused on biomechanical changes in posture control, while secondary outcomes included subjective/objective fatigue assessments. Bonferroni correction addressed multiple comparisons (α = 0.05/number of tests) [25].

## 3. Results

### 3.1. Changes in Objective and Subjective Indices of Mental Fatigue at Different Intervention Stages

Figure 4 illustrates the fatigue intervention effects on psychomotor vigilance (refers to an individual’s ability to maintain attention and rapid response, measured through PVT tasks) and subjective states. Post-intervention PVT response times increased significantly versus baseline (*p* < 0.05), persisting throughout the measurements (Figure 4a), indicating sustained cognitive impairment. Subjective fatigue peaked post-intervention, partially recovered but remained elevated versus baseline (Figure 4b). Despite reward-incentivized design, motivation decreased significantly yet maintained moderate levels (Figure 4c). Effort perception surged post-fatigue (*p* < 0.001) and remained elevated despite partial fatigue recovery (Figure 4d), demonstrating dissociation between objective fatigue decay and persistent effort awareness. The intensity of this mental fatigue intervention rapidly induced a heightened perception of effort in individuals. In the short term, this perception of effort did not diminish alongside the subjective and objective perception of fatigue, remaining consistently high.

### 3.2. Alterations in Time-Domain Indicators of Upright Stability Under Various Sensory Interference Conditions with Mental Fatigue

The findings related to the time-domain indicators in the AP and medial–lateral (ML) directions of human upright stability, both pre- (NORMAL) and post-fatigue intervention (MF), under differing sensory input conditions, are presented in Table 2 and Table 3. All data in this section satisfy normality requirements and were analyzed using paired *t*-tests.

#### 3.2.1. Alterations in Time-Domain Indicators of Anterior–Posterior Upright Stability Under Fatigue

The velocity and total displacement of the COP in the AP direction exhibited an increase before and after intervention; however, no statistically significant differences were observed. Notably, the sway index demonstrated statistically significant differences before and after fatigue under conditions devoid of proprioceptive interference and visual occlusion interference (*p* < 0.05), with fatigue leading to an increase in the AP direction sway index under open-eyes (O-H) conditions. The results are presented in Table 2.

#### 3.2.2. Alterations in Time-Domain Indicators of Upright Stability in Internal and External Directions Under Fatigue

No statistically significant differences were observed in the ML direction velocity and total displacement length of the COP before and after fatigue. However, a statistically significant difference was noted in the ML direction sway index of COP before and after fatigue under conditions devoid of proprioceptive interference and visual occlusion (*p* < 0.05). Fatigue led to an increase in the sway index in the ML direction under these standard conditions. The results are presented in Table 3.

#### 3.2.3. Alterations in Overall Time-Domain Indicators of Upright Stability Under Fatigue

Mental fatigue led to a notable increase in the COP movement area under closed-eye conditions (both hard and soft surfaces), with significant statistical differences observed (*p* < 0.001 for hard surface; *p* < 0.01 for soft surface). However, under open-eye conditions, the increase in COP movement area was less pronounced and only reached significance for the hard surface condition (*p* < 0.05). Additionally, the unit area trajectory length significantly shortened under mental fatigue, particularly for closed-eye conditions (*p*< 0.001 for hard surface; *p* < 0.01 for soft surface), suggesting a more constrained movement pattern (Table 4).

### 3.3. Changes in Frequency-Domain Energy Indicators of Upright Stability Under Different Sensory Interference Conditions Under Mental Fatigue

#### 3.3.1. Alterations in Time-Domain Indicators of Upright Stability During Fatigue

The dataset exhibited normal distribution and underwent three-way repeated measures ANOVA to evaluate the main and interactive effects of mental fatigue (pre/post-intervention), visual occlusion, and proprioceptive interference on energy proportions (E_Vision, E_Vestibule, E_Cerebellum, E_Proprioception) across four sensory-related frequency bands (Table 5). Main effects analysis revealed that mental fatigue significantly influenced cerebellar (*p* < 0.01) and proprioceptive (*p* < 0.001) energy dynamics. Visual occlusion exerted a dominant effect on visual, cerebellar, and proprioceptive bands (all *p* < 0.001), while proprioceptive interference selectively impacted the proprioceptive band (*p* < 0.01). Interaction analysis identified no significant three-way interactions. However, mental fatigue × visual occlusion interactions affected visual, cerebellar, and proprioceptive energy (*p* < 0.05), and visual occlusion × proprioceptive interference interactions modulated proprioceptive energy (*p* < 0.01).

#### 3.3.2. Results of Simple Effect Analysis on Multi-Sensory Corresponding Frequency-Domain Energy Ratio

Fluctuations in visual index values were noted both before and after fatigue; however, no significant statistical differences were identified under any of the conditions. Mental fatigue did not have a significant independent impact on the energy of visual information input frequency bands. Under O-S conditions, there was a statistically significant increase in the proportion of vestibular sense corresponding energy post-fatigue (*p* < 0.05). In contrast, under C-S conditions, the cerebellar frequency band energy proportion significantly decreased post-fatigue, exhibiting a high level of statistical significance (*p* < 0.001). Similarly, under both C-H and C-S conditions, the proprioceptive energy frequency proportion significantly decreased post-fatigue, again with a high level of statistical significance (*p* < 0.001). Finally, under O-H conditions, the proprioceptive energy frequency proportion also decreased post-fatigue, reaching statistical significance (*p* < 0.05). These results are detailed in Table 6 and Figure 5.

#### 3.3.3. The Impact of Visual Occlusion and Proprioceptive Interference on the Energy Proportion of the Multi-Sensory Frequency-Domain

A comparative analysis from a visual occlusion perspective revealed that prior to fatigue induction, the energy proportion index corresponding to the visual frequency band significantly decreased after visual occlusion, regardless of proprioceptive interference (*p* < 0.001). Concerning post-fatigue induction, under non-proprioceptive interference conditions, the energy index corresponding to visual occlusion significantly decreased (*p* < 0.05), with a reduced decrease relative to pre-fatigue levels. Under proprioceptive interference conditions, the decrease in the energy proportion index corresponding to the visual frequency band due to visual occlusion further diminished, with no significant statistical difference observed. Prior to fatigue induction, the energy proportion index corresponding to the vestibular sense frequency band increased after visual occlusion, regardless of proprioceptive interference, with no significant statistical difference observed. However, the compensatory role of the vestibular sense for visual input cannot be discounted. For post-fatigue induction, no significant change in this index was observed. By examining the values, it was found that the energy proportion index of the cerebellar frequency band significantly increased before and after visual occlusion under various conditions, except under fatigue and proprioceptive disturbance conditions where the increase was relatively less, but still statistically significant (*p* < 0.05). Under other conditions, the cerebellar energy index rose significantly (*p* < 0.001). Visual occlusion under all conditions resulted in a significant increase in the energy proportion index. See Table 7.

#### 3.3.4. The Influence of Proprioceptive Interference Conditions on the Energy Proportion of the Multi-Sensory Frequency-Domain

A comparative analysis from the proprioceptive interference perspective reveals no statistically significant differences (*p* > 0.05) in the values of the other three frequency bands before and after proprioceptive interference under each condition, with no noticeable numerical variations. However, under the visual occlusion condition, a significant change in energy proportion is observed before and after proprioceptive intervention (*p* < 0.05). This is illustrated in Table 8.

### 3.4. Examination of COP Signal in Various Sensory Interference Conditions During Mental Fatigue

Compared to pre-fatigue levels, the sample entropy index of subjects experiencing mental fatigue significantly decreased, as illustrated in Table 9. Under both closed-eye and open-eye hard ground conditions, the onset of mental fatigue led to a highly significant decrease in the sample entropy index in both directions (*p* < 0.003). This clearly suggests that under these mental fatigue conditions, the posture control system is trending toward regularization and automation. However, under open-eyes and proprioception interference conditions, the decrease in sample entropy due to mental fatigue was no longer significant (*p* > 0.05), as shown in Table 9.

## 4. Discussion

This study investigated the effects of mental fatigue on postural control under various sensory interference conditions using biomechanical data derived from COP metrics and advanced nonlinear methods, including sample entropy (SampEn) analysis and frequency-domain energy decomposition via DWT (SampEn is a measure of signal complexity. Lower SampEn values indicate more regular and predictable patterns, while higher values suggest greater variability. In our study, SampEn was used to assess the complexity of postural control signals under different conditions). The findings highlight the complex interactions between mental fatigue and multi-sensory integration, revealing critical insights into compensatory mechanisms at the biomechanical and neural levels.

### 4.1. Sample Entropy Variability: Mechanistic Insights

The observed changes in SampEn reflect alterations in postural control complexity under mental fatigue. SampEn measures the predictability and regularity of COP signals, with higher values indicating greater variability—a marker for adaptive capacity in postural control. Our results demonstrate a significant reduction in SampEn values in certain sensory conditions, suggesting that mental fatigue impairs the system’s ability to respond flexibly to perturbations. Mechanistically, this reduced variability likely stems from mental fatigue’s interference with higher-order neural processes involved in sensory integration. Specifically, the existing literature indicates that fatigue may impair cortical and subcortical communication [26], such as between the prefrontal cortex and cerebellum, which are critical for maintaining dynamic adjustments in posture [27]. The decrease in SampEn under proprioceptive interference (O-S and C-S conditions) suggests that when proprioceptive inputs are compromised, mental fatigue exacerbates the dependence on other sensory systems (e.g., vestibular or visual). This aligns with findings from prior studies, where impaired proprioception magnified reliance on alternate sensory resources to maintain stability [26]. However, under certain conditions, such as open surface (O-H), SampEn did not significantly change despite mental fatigue. This differential response highlights the necessity of task-specific and context-dependent analyses to fully understand the neural and biomechanical basis of postural control under fatigue.

### 4.2. Integration with Biomechanical Data

From a biomechanical perspective, the significant increase in COP velocity, accompanied by decreased SampEn, under C-H and O-S conditions suggests that mental fatigue disrupts the coordination mechanism governing postural stability. The enlarged COP area under these conditions aligns with biomechanical theories positing that increased sway may reflect reduced stability or diminished sensorimotor precision. Correspondingly, the decline in proprioceptive frequency band energy proportions observed with DWT further supports the hypothesis that fatigue compromises proprioceptive regulation. While traditional sway indicators (e.g., COP displacement and velocity) provide foundational biomechanical insights, this study’s integration of entropy and frequency-domain analysis highlights novel dynamics in sensory compensation. The significant energy redistribution toward vestibular and cerebellar inputs in fatigued states reinforces the role of these systems in compensating for compromised proprioception under cognitive load—theoretically supporting models where fatigue modifies sensory reweighting to stabilize posture.

### 4.3. Practical Implications of Datasets

These findings have some implications for practical use. For example, the finding of a decrease in postural control under a dual interfered condition (C-S) indicates that people who perform challenging cognitive tasks in environments with compromised sensory information (e.g., low visibility or uneven surface in workplaces) are more likely to experience instability or fall. The predictable decline of proprioceptive effectiveness with fatigue indicates that COP-based sensors can be used for real-time assessment of proprioceptive function in rehabilitation or ergonomic evaluations. The results of SampEn can be used to develop adaptive balance training programs (e.g., proprioceptive training, or cognitive dual-task training) to counteract the accumulative effect of fatigue on sensory–motor control. In clinical applications, entropy-based metrics can serve as sensitive indicators to evaluate the recovery progress of populations susceptible to mental fatigue (e.g., older adults, patients with neurological disorders).

### 4.4. Comprehensive Synthesis of Findings

Taken together, our results suggest that mental fatigue selectively impairs proprioceptive information processing, while the visual and vestibular systems are recruited to a greater or lesser extent depending on the task constraints and the interference condition. Such hierarchical sensory prioritization is consistent with recent theories of multi-sensory reweighting in balance tasks but indicates that mental fatigue leads to faster changes in sensory reliance. From a methodological point of view, this study extends the current state of biomechanical research by integrating traditional measures of COP sway with nonlinear (entropy measures) and spectral (DWT) analyses of the sway data. This multifaceted approach offers a more complete picture of postural control, linking static measures with dynamic compensatory mechanisms. Finally, in contrast to some previous studies, our data suggest that fatigue-related instability can occur without large center of pressure displacements but is instead reflected in changes in sensorimotor complexity. These results highlight the importance of going beyond traditional metrics and encourage the use of more complex analytical tools in the study of fatigue.

### 4.5. Explaining Fundamental Principles and Applications

The theoretical background of this study is based on the multi-sensory integration (visual, proprioceptive, vestibular) and neural resource allocation required for dynamic postural control [28]. Mental fatigue is assumed to affect this multi-sensory integration and neural resource allocation. High cognitive loads further aggravate this situation due to the competition for attentional resources, which decreases the efficiency of sensory reweighting, particularly when multi-sensory integration is required [29]. Potential applications of the proposed approach are not limited to an experimental setting but could also be envisioned in different fields of application. Wearable sensor technologies that allow for real-time computation of DWT and SampEn could help to detect early signs of instability induced by fatigue in high-risk situations [30] such as aviation, sports, or industrial settings. Individualized feedback from such a system might allow for the dynamic adjustment of task constraints (e.g., workload modulation, surface adjustment) to reduce the likelihood of accidents resulting from impaired postural control. In addition, the integration of entropy and energy measures into rehabilitation training could help to improve both diagnostic and training tools for neuromuscular or cognitive impairments.

### 4.6. Limitations

(1) Participant Selection Bias: This study exclusively recruited healthy young males as participants, which limits the generalizability of the findings. The results cannot be directly extended to other populations, such as females, older adults, and individuals with specific health conditions. Furthermore, the study did not evaluate the posture control dynamics of these particular groups under mental fatigue intervention. Future research should include a broader sample demographic to enhance the external validity of the findings.

(2) Experimental Task and Context Constraints: The experimental setup used highly controlled laboratory environments to induce mental fatigue and test postural control under various interference conditions. However, these controlled settings may not accurately reflect real-world conditions, where sensory disturbances are often more complex and dynamic (e.g., uneven surfaces, crowded environments). This limits the ecological validity of the findings, as the results may not fully represent the effects of fatigue on postural control in practical applications.

(3) Methodological Limitations: While this study used DWT to analyze changes in frequency-domain energy proportions and sample entropy, these metrics have limitations in capturing the full complexity of postural control mechanisms. Additionally, the non-significant changes in traditional time-domain sway indicators suggest their limited sensitivity to the effects of mental fatigue on postural control. Future studies should employ a more comprehensive methodological approach, such as multimodal data integration, to better characterize fatigue-induced changes in posture control.

(4) Lack of Mechanistic Exploration: Although this study hypothesized that mental fatigue affects proprioceptive input modulation and cerebellar energy distribution, the underlying neural mechanisms were not directly investigated. The conclusions are largely based on statistical analyses and literature references without direct evidence from neurophysiological data. Future research could employ neuroimaging techniques (e.g., fMRI, EEG) or electrophysiological recordings to uncover the neural pathways involved in mental fatigue and its impact on multi-sensory integration.

### 4.7. Contributions to the Field

This study advances the understanding of mental fatigue through three key contributions.

(1)Mechanistic differentiation: By isolating mental fatigue from physical fatigue, we reveal its unique disruption of multi-sensory integration in postural control, refining cognitive load theory.(2)Analytical innovation: The integration of DWT and SampEn with traditional COP metrics establishes a novel framework to quantify fatigue-induced sensory reorganization.(3)Sensory compensation dynamics: We demonstrate fatigue-driven proprioceptive decline with compensatory vestibular–visual shifts, advancing sensory reweighting theories in dynamic environments.(4)Practical applications: Findings inform sensor-based fatigue monitoring systems for aviation/sports safety and targeted rehabilitation protocols for vulnerable populations.(5)Ergonomic implications: The results support cognitive–physical integrated training designs to mitigate fatigue-related instability in occupational/rehabilitation contexts.

### 4.8. Future Directions and Practical Implications

This study lays the groundwork for understanding the variability of postural control and multi-sensory compensation mechanisms under mental fatigue. Future research should incorporate real-world scenarios (e.g., astronaut training, fatigue monitoring during driving) and evaluate a more diverse demographic (e.g., older adults or patients with chronic illnesses) to validate these findings in practical applications. Moreover, integrating experimental designs with neuroimaging techniques could elucidate the specific neural mechanisms of sensory integration under fatigue, providing critical insights for preventing fatigue-induced instability during postural control. Furthermore, static balance pertains to the human body’s ability to maintain stability by integrating sensory inputs such as visual, vestibular, and proprioceptive sensations while remaining stationary. This serves as a critical metric for evaluating postural control and balance capabilities and is extensively employed in health assessments and rehabilitation medicine. Although this study primarily centers on dynamic balance, understanding static balance offers valuable contextual insights for the research. For example, the observed increase in the amplitude of subject sway under closed-eye conditions mirrors the closed-eye test used in static balance evaluations. Future studies could further investigate the application of static balance across diverse populations and its relevance in postural control assessments.

## 5. Conclusions

This study successfully induced mental fatigue through cognitive intervention methods and investigated its impact on postural control capability and sensory compensation mechanisms. Using inertial platform testing and COP signal analysis, the following key findings were obtained:

### 5.1. Overall Impact of Mental Fatigue on Postural Control

Although traditional time-domain sway indicators showed no significant changes under the influence of mental fatigue, significant differences in sample entropy and frequency band energy proportions suggest that fatigue negatively impacts postural control. Specifically, the reduction in proprioceptive energy proportions indicates that mental fatigue may impair proprioceptive regulation during postural maintenance.

### 5.2. Adaptive Adjustments in Multi-Sensory Compensation Mechanisms

The findings demonstrate that under mental fatigue, different sensory inputs exhibit adaptive compensation mechanisms. For instance, the increased energy proportion of vestibular-related frequency bands suggests that vestibular inputs may compensate for impaired proprioceptive function, maintaining postural stability. This compensation highlights the importance of dynamic collaboration among multiple sensory systems, though the specific mechanisms require further investigation.

### 5.3. Significant Changes in Specific Experimental Conditions

Under certain interference conditions (e.g., C-H, C-S), proprioceptive and cerebellar frequency band energy proportions significantly decreased, suggesting that mental fatigue may directly affect the central nervous system’s ability to integrate sensory information. Additionally, visual input reliance remained consistent across all conditions, implying that mental fatigue primarily impacts mechanisms related to proprioception and vestibular regulation rather than visual processing.

## Figures and Tables

**Figure 1 sensors-25-01470-f001:**
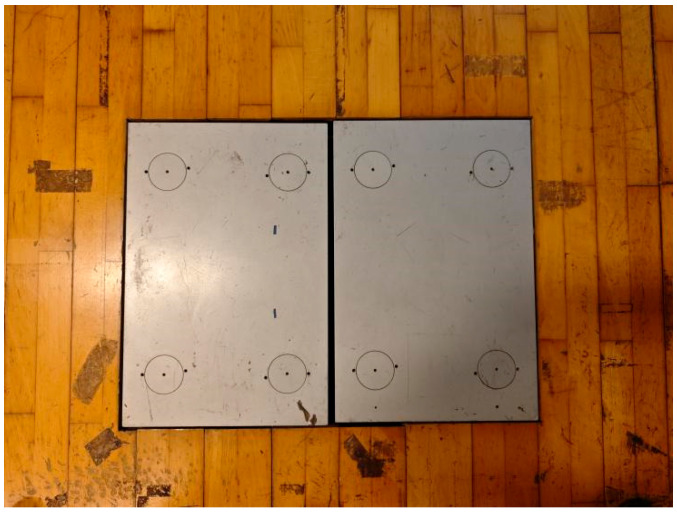
Kistler dynamometer.

**Figure 2 sensors-25-01470-f002:**
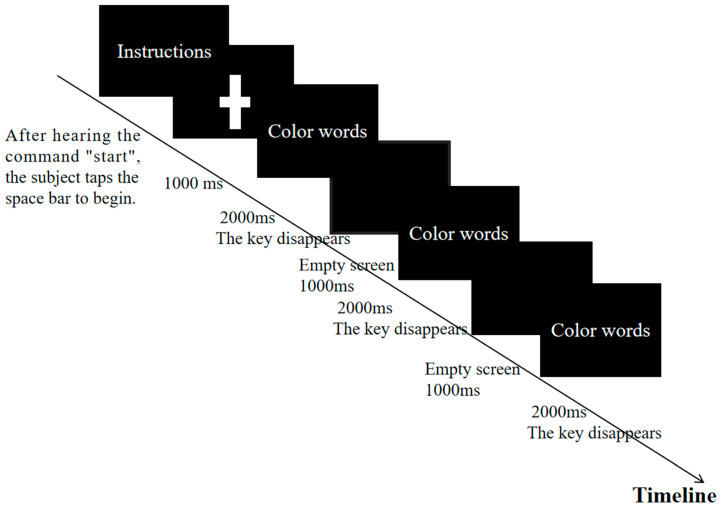
Stroop flow diagram.

**Figure 3 sensors-25-01470-f003:**
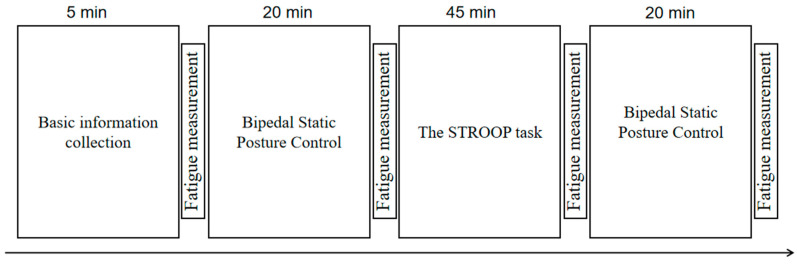
Experimental flow chart.

**Figure 4 sensors-25-01470-f004:**
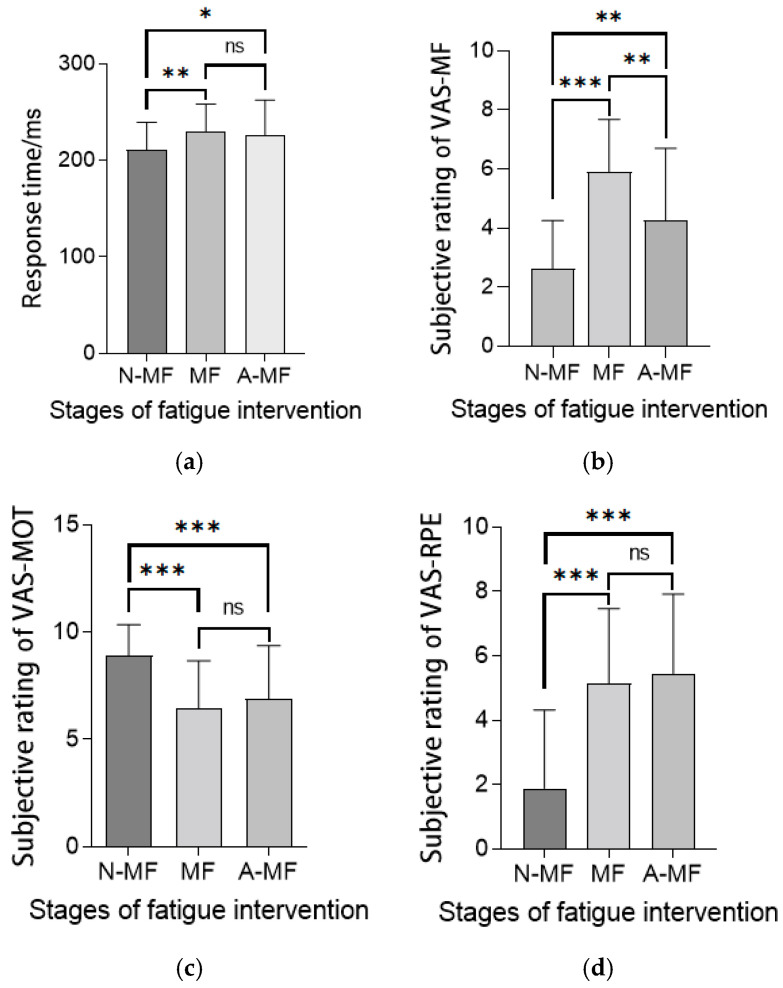
(**a**) Response time of PVT. (**b**) Subjective rating of VAS-MF. (**c**) Subjective rating of VAS-MOT. (**d**) Subjective rating of VAS-RPE. Various indicators at different stages of fatigue intervention. Note: Statistical differences in different fatigue stages are represented as * *p* < 0.05, ** *p* < 0.01, *** *p* < 0.001. Pre-intervention fatigue stage is natural state (normal state), denoted as N-MF; immediately after mental fatigue intervention is fatigue state (mental fatigue state), denoted as MF; 20 min post-fatigue intervention is post-fatigue state, denoted as A-MF. ns is not significant.

**Figure 5 sensors-25-01470-f005:**
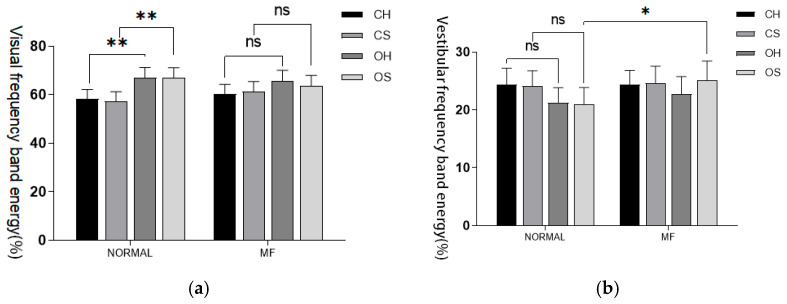
(**a**) Visual input frequency band. (**b**) Vestibular sensory input frequency band. (**c**) Cerebellar input frequency band. (**d**) Proprioceptive input frequency band. Proportion of energy in each sensory input frequency band before and after fatigue intervention. Note: Statistical differences in different fatigue stages are represented as * *p* < 0.05, ** *p* < 0.01, *** *p* < 0.001. ns is not significant.

**Table 1 sensors-25-01470-t001:** Basic information of subjects.

Sample Size	Age (Years)	Height (cm)	Weight (kg)	BMI (kg/m^2^)	PQSI (Points)
N = 27	20.30 ± 1.86	176.04 ± 5.60	69.76 ± 10.06	22.50 ± 3.02	4.74 ± 2.01

Note: BMI (Body Mass Index); PQSI (Pittsburgh Sleep Quality Index).

**Table 2 sensors-25-01470-t002:** Results of AP direction upright stability indicators under different interference conditions before and after mental fatigue intervention.

Indicators	Interference Conditions	Normal	MF	*p*
Velocity_AP(mm/s)	C-H	54.05 ± 8.82	54.13 ± 8.62	0.865
	C-S	55.01 ± 8.96	55.92 ± 8.88	0.107
	O-H	53.18 ± 8.96	53.62 ± 8.68	0.443
	O-S	53.12 ± 9.20	53.55 ± 8.59	0.386
Length_AP(mm)	C-H	3242.60 ± 529.35	3247.52 ± 517.39	0.865
	C-S	3299.76 ± 537.76	3354.37 ± 532.63	0.107
	O-H	3189.98 ± 537.51	3216.50 ± 520.68	0.443
	O-S	3186.66 ± 551.76	3212.72 ± 515.34	0.386
DI_AP(%)	C-H	77.62 ± 2.61	77.82 ± 2.98	0.345
	C-S	79.88 ± 2.48	80.04 ± 1.90	0.472
	O-H	76.57 ± 2.88	77.17 ± 2.99 *	0.030
	O-S	79.10 ± 2.45	79.32 ± 2.61	0.439

Note: Statistical differences in different fatigue stages are represented as * *p* < 0.05.

**Table 3 sensors-25-01470-t003:** Results of time-domain indicators for ML direction upright stability under different interference conditions before and after mental fatigue intervention.

Indicators	Interference Conditions	Normal	MF	*p*
Velocity_ML(mm/s)	C-H	33.99 ± 6.76	33.70 ± 6.46	0.385
	C-S	31.75 ± 6.50	31.92 ± 5.79	0.598
	O-H	34.78 ± 7.27	34.32 ± 7.02	0.307
	O-S	31.49 ± 6.63	31.45 ± 6.10	0.918
Length_ML(mm)	C-H	2039.20 ± 405.44	2021.46 ± 387.49	0.385
	C-S	1904.45 ± 389.73	1914.79 ± 347.17	0.598
	O-H	2086.75 ± 436.34	2058.63 ± 421.34	0.307
	O-S	1889.22 ± 397.57	1886.93 ± 366.07	0.918
DI_ML(%)	C-H	48.55 ± 3.14	48.24 ± 3.55	0.243
	C-S	45.82 ± 3.01	45.53 ± 2.35	0.304
	O-H	49.78 ± 3.39	49.08 ± 3.52 *	0.033
	O-S	46.67 ± 2.98	46.38 ± 3.07	0.383

Note: Statistical differences in different fatigue stages are represented as * *p* < 0.05.

**Table 4 sensors-25-01470-t004:** Results of overall time-domain indicators of upright stability under different interference conditions before and after mental fatigue intervention.

Indicators	Interference Conditions	Normal	MF	*p*
Velocity(mm/s)	C-H	69.76 ± 11.80	69.65 ± 11.28	0.854
	C-S	68.98 ± 11.71	69.92 ± 11.29	0.138
	O-H	69.59 ± 12.23	69.63 ± 11.76	0.953
	O-S	67.25 ± 12.08	67.61 ± 11.11	0.543
Area(mm²)	C-H	240.93 ± 428.95	274.15 ± 323.19 ***	0.001
	C-S	361.23 ± 374.40	468.82 ± 391.35 **	0.006
	O-H	179.42 ± 251.90	238.91 ± 303.25 *	0.016
	O-S	319.83 ± 297.70	437.83 ± 525.01	0.140
Length_per_area (mm)	C-H	53.73 ± 56.84	38.25 ± 37.60 ***	<0.001
	C-S	22.30 ± 16.58	16.22 ± 12.91 **	0.002
	O-H	64.41 ± 76.52	41.47 ± 33.18 **	0.003
	O-S	24.68 ± 21.96	23.76 ± 24.52	0.447

Note: Statistical differences in different fatigue stages are represented as * *p* < 0.05, ** *p* < 0.01, *** *p* < 0.001.

**Table 5 sensors-25-01470-t005:** Results of main and interactive effects analysis on energy proportion indicators of each frequency band under different interference conditions.

Interference Conditions	E_Vision		E_Vestibule		E_Cerebellum		E_Proprioception	
	*F*	*p*	*F*	*p*	*F*	*p*	*F*	*p*
Mental fatigue	0.072	0.789	2.293	0.134	8.976 **	0.004	18.819 ***	<0.001
Visual obscuration	23.047 ***	<0.001	3.069	0.084	56.681 ***	<0.001	39.232 ***	<0.001
Body interference	0.138	0.711	0.235	0.629	0.062	0.804	7.010 **	0.010
Fatigue * Vision	5.658 *	0.020	2.570	0.113	6.721 **	0.011	4.040	0.048
Fatigue * body	0.014	0.905	0.841	0.362	1.422	0.237	0.098	0.755
Visual * ontology	0.200	0.656	0.628	0.431	0.065	0.800	10.540 **	0.002
Fatigue * Visual * Ontology	0.690	0.409	0.432	0.513	0.493	0.484	2.264	0.136

Note: Statistical differences in different fatigue stages are represented as * *p* < 0.05, ** *p* < 0.01, *** *p* < 0.001.

**Table 6 sensors-25-01470-t006:** Results of multi-sensory corresponding frequency band energy proportion before and after mental fatigue intervention.

Indicators	Interference Conditions	Normal	MF	*p* Value
E_Vision (%)	C-H	58.14 ± 18.29	60.42 ± 17.45	0.342
	C-S	57.29 ± 18.20	61.23 ± 19.29	0.098
	O-H	67.10 ± 19.09	65.80 ± 19.51	0.559
	O-S	67.13 ± 18.26	63.65 ± 19.39	0.181
E_Vestibule (%)	C-H	24.37 ± 12.97	24.44 ± 10.98	0.964
	C-S	24.10 ± 12.18	24.60 ± 13.56	0.760
	O-H	21.24 ± 11.86	22.76 ± 13.64	0.390
	O-S	21.04 ± 12.92	25.12 ± 15.12 *	0.035
E_Cerebellum (%)	C-H	16.01 ± 10.26	14.00 ± 9.98	0.101
	C-S	16.63 ± 10.66	12.72 ± 9.70	<0.001
	O-H	10.59 ± 10.28	10.61 ± 8.04	0.974
	O-S	10.80 ± 8.75	10.32 ± 7.17	0.608
E_Proprioception (%)	C-H	1.48 ± 1.49	1.13 ± 1.20 ***	0.001
	C-S	1.98 ± 2.10	1.45 ± 1.78 ***	0.001
	O-H	1.08 ± 1.28	0.83 ± 0.99 *	0.026
	O-S	1.04 ± 1.23	0.91 ± 0.95	0.198

Note: Statistical differences in different fatigue stages are represented as * *p* < 0.05, *** *p* < 0.001.

**Table 7 sensors-25-01470-t007:** Results of multi-sensory corresponding frequency band energy proportion under different visual masking interference conditions.

Indicators	Interference Conditions	NORMAL	MF	*p* Value
E_Vision (%)	H	67.10 ± 19.09	58.14 ± 18.29 ***	0.001
	S	67.13 ± 18.26	57.29 ± 18.20 ***	<0.001
	MF-H	65.80 ± 19.51	60.42 ± 17.45 *	0.015
	MF-S	63.65 ± 19.39	61.23 ± 19.29	0.370
E_Vestibule (%)	H	21.24 ± 11.86	24.37 ± 12.97	0.086
	S	21.04 ± 12.92	24.10 ± 12.18	0.053
	MF-H	22.76 ± 13.64	24.44 ± 10.98	0.277
	MF-S	25.12 ± 15.12	24.60 ± 13.56	0.783
E_Cerebellum (%)	H	10.59 ± 10.28	16.01 ± 10.26 ***	<0.001
	S	10.80 ± 8.75	16.63 ± 10.66 ***	<0.001
	MF-H	10.61 ± 8.04	14.00 ± 9.98 ***	<0.001
	MF-S	10.32 ± 7.17	12.72 ± 9.70	0.028
E_Proprioception (%)	H	1.08 ± 1.28	1.48 ± 1.49 ***	0.001
	S	1.04 ± 1.23	1.98 ± 2.10 ***	<0.001
	MF-H	0.83 ± 0.99	1.13 ± 1.20 **	0.003
	MF-S	0.91 ± 0.95	1.45 ± 1.78 ***	<0.001

Note: Statistical differences in different fatigue stages are represented as * *p* < 0.05, ** *p* < 0.01, *** *p* < 0.001.

**Table 8 sensors-25-01470-t008:** Results of multi-sensory corresponding frequency band energy proportion under different proprioceptive interference conditions.

Indicators	Interference Conditions	Normal	MF	*p* Value
E_Vision (%)	C	24.37 ± 12.97	24.10 ± 12.18	0.749
	O	21.24 ± 11.86	21.04 ± 12.92	0.991
	MF-C	24.44 ± 10.98	24.60 ± 13.56	0.723
	MF-O	22.76 ± 13.64	25.12 ± 15.12	0.348
E_Vestibule (%)	C	16.01 ± 10.26	16.63 ± 10.66	0.881
	O	10.59 ± 10.28	10.80 ± 8.75	0.906
	MF-C	14.00 ± 9.98	12.72 ± 9.70	0.919
	MF-O	10.61 ± 8.04	10.32 ± 7.17	0.183
E_Cerebellum (%)	C	1.48 ± 1.49	1.98 ± 2.10	0.665
	O	1.08 ± 1.28	1.04 ± 1.23	0.853
	MF-C	1.13 ± 1.20	1.45 ± 1.78	0.265
	MF-O	0.83 ± 0.99	0.91 ± 0.95	0.742
E_Proprioception (%)	C	24.37 ± 12.97	24.10 ± 12.18 **	0.002
	O	21.24 ± 11.86	21.04 ± 12.92	0.707
	MF-C	24.44 ± 10.98	24.60 ± 13.56 *	0.017
	MF-O	22.76 ± 13.64	25.12 ± 15.12	0.420

Note: Statistical differences in different fatigue stages are represented as * *p* < 0.05, ** *p* < 0.01.

**Table 9 sensors-25-01470-t009:** The results of the sample entropy of the COP signal on the foot under different proprioceptive interference conditions.

Indicators	Interference Conditions	Normal	MF	*p* Value
SampEn_AP	C-H	0.49 ± 0.26	0.40 ± 0.22 ***	<0.001
	C-S	0.31 ± 0.15	0.25 ± 0.13 ***	<0.001
	O-H	0.51 ± 0.30	0.42 ± 0.22 **	0.003
	O-S	0.31 ± 0.17	0.29 ± 0.18	0.448
SampEn_ML	C-H	0.87 ± 0.46	0.73 ± 0.50 ***	<0.001
	C-S	0.57 ± 0.32	0.48 ± 0.31 ***	0.001
	O-H	0.91 ± 0.46	0.78 ± 0.46 ***	0.001
	O-S	0.60 ± 0.35	0.55 ± 0.42	0.227

Note: Statistical differences in different fatigue stages are represented as ** *p* < 0.01, *** *p* < 0.001.

## Data Availability

Due to the fact that all participants are students, in order to protect privacy, this study does not disclose the original data. This study can provide the data; you can contact the corresponding author email: fengran@cqu.edu.cn.

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
