# Peer review of "Sensor-Based Assessment of Mental Fatigue Effects on Postural Stability and Multi-Sensory Integration"

_sensors, 2025, doi:10.3390/s25051470_

Round 1

Reviewer 1 Report (New Reviewer)

Comments and Suggestions for Authors

This is an excessively detailed study assessing the effects of fatigue on “postural stability and multi-sensory integration”, as the authors have stated.  The abstract itself is overly detailed (40 lines long) and as a result, it is difficult to follow the authors’ experimental design, and the intention of  what the paper is aiming to do.  

The details of this manuscript are so excessive that the experimental protocol is markedly difficult to follow.  The introduction itself is 100 lines long.

The participants they recruited were all males.  One wonders why females weren’t included in their cohort. 

Beginning on line 120, the experimental detail is outlined and described introducing three independent variables of postural performance.  This small section alone goes on for 20 lines, when it could have been outlined with a single reference referencing Computerized Dynamic Posturography, a validated clinical tool .  Again the detailed English explanation (which is not at all streamlined in any way) prevents the reader from understanding their protocol, and makes some of the discussion almost incomprehensible.  The mathematical statistics section also goes on for 20 lines and is almost incomprehensible.  The English is technically correct grammatically but its syntax is poor and makes it extremely difficult to read. 

The results section beginning on line 324 is also overly detailed and the poor English syntax again makes it difficult to follow.  On line 332 they use the term “psychomotor vigilance” but did not define this term or discuss what they were trying to express.  Some of the other terminology is used inappropriately. For example online 362 they describe “diverse sensory input conditions” which is not an accurate way in English to express the experimental conditions.  The term “differing” would be more accurate. 

Many of the tables (especially table 5) are virtually incomprehensible.

Beginning online for 25, the authors discuss “the influence of the fatigue and interference conditions on the distribution of multi sensory energy”.  Again this is a term that is not discussed and not defined.

The 160 word discussion is also far too lengthy and difficult to understand.  I am not familiar with the term “sample entropy” in line 509 and this needs to be discussed in plain English language using proper syntax.  In the table at the end they do define the term entropy, as understood by physicists, but the concept of “entropy” as studied by physicists is not really clinically relevant. 

 On line 516 they make comments about “what existing literature indicates” but there are no references supplied. Similarly, on line 521 they comment that their findings “aligns with findings from prior studies” but supply no references.

On line 542, they make the statements that their study “offers actionable insights for practical applications”.  I assume they are attempting to make the statement that their findings are clinically relevant, but again the syntax used is difficult to understand.

In the first part of section 4.4, their “comprehensive synthesis of findings” is reasonably well written but far too detailed.  On line 573, they stated their findings are “consistent with central resource theory”.  I am not familiar with this term and there is no discussion of it or a reference for it.

The “limitations” section is reasonably well written.  However their “contributions to the field” section is far too run on.

In summary this manuscript is far too detailed to be a reasonable contribution to the clinical literature. It is also written using very poor, rigid formalized English syntax, and this makes it harder to extract their findings, or what they were trying to show.    

Comments on the Quality of English Language

Your English syntax is poor and your writing is far too detailed to be understood by clinicians detailed.  Clinical studies should not be written in formalized English and all terms must be described in understandable ways.

Author Response

Dear REVIEWER 1

Thank you very much for your insightful comments and suggestions on our manuscript. We greatly appreciate your time and effort in reviewing our work, and we found your feedback to be very constructive.

1.This is an excessively detailed study assessing the effects of fatigue on “postural stability and multi-sensory integration”, as the authors have stated.  The abstract itself is overly detailed (40 lines long) and as a result, it is difficult to follow the authors’ experimental design, and the intention of  what the paper is aiming to do.  

ANSWER: Thank you for your review and suggestion, we have simplified the abstract section as per your suggestion.

2,The details of this manuscript are so excessive that the experimental protocol is markedly difficult to follow.  The introduction itself is 100 lines long.

ANSWER: Thank you for your review and suggestion, we will simplify the introduction part to make it more concise, highlighting the background and purpose of the study. We will eliminate redundant information to ensure that readers can easily understand our experimental design and research objectives.

3.The participants they recruited were all males.  One wonders why females weren’t included in their cohort.

ANSWER: Thank you for your review and suggestion, we acknowledge that the study only included male participants. We will be explicit in the text about why we chose to recruit only male participants, primarily to control for the effects of gender differences on postural control and cognitive fatigue. Also, we will highlight the importance of including a more diverse group of participants in future studies.

4.Beginning on line 120, the experimental detail is outlined and described introducing three independent variables of postural performance.  This small section alone goes on for 20 lines, when it could have been outlined with a single reference referencing Computerized Dynamic Posturography, a validated clinical tool .  Again the detailed English explanation (which is not at all streamlined in any way) prevents the reader from understanding their protocol, and makes some of the discussion almost incomprehensible.  The mathematical statistics section also goes on for 20 lines and is almost incomprehensible.  The English is technically correct grammatically but its syntax is poor and makes it extremely difficult to read.

ANSWER: Thank you for your review and suggestions. We have deleted the part you mentioned in the introduction and simplified the statistics and polished English.

5.The results section beginning on line 324 is also overly detailed and the poor English syntax again makes it difficult to follow.  On line 332 they use the term “psychomotor vigilance” but did not define this term or discuss what they were trying to express.  Some of the other terminology is used inappropriately. For example online 362 they describe “diverse sensory input conditions” which is not an accurate way in English to express the experimental conditions.  The term “differing” would be more accurate.

ANSWER: We will only describe the significant results in detail (p <0.05) and remove the lengthy description of the non-significant results (p> 0.05). We will clearly define the term "psychomotor alertness" and ensure that all terms are used appropriately.

6.Many of the tables (especially table 5) are virtually incomprehensible.

ANSWER: Thank you for your review and suggestions. We have made some adjustments to the table and added explanations to the part mentioned in the results to help us understand Table 5.

7.Beginning online for 25, the authors discuss “the influence of the fatigue and interference conditions on the distribution of multi sensory energy”.  Again this is a term that is not discussed and not defined.

ANSWER: Thank you for your review, which has been modified to the Energy Proportion of Multi-Sensory Frequency Domain and has been mentioned in 2.2.2- (4).

8.The 160 word discussion is also far too lengthy and difficult to understand.  I am not familiar with the term “sample entropy” in line 509 and this needs to be discussed in plain English language using proper syntax.  In the table at the end they do define the term entropy, as understood by physicists, but the concept of “entropy” as studied by physicists is not really clinically relevant.

ANSWER:Thank you for your review. "Sample entropy (SampEn) is a measure of signal complexity. Lower SampEn values indicate more regular and predictable patterns, while higher values suggest greater variability. In our study, SampEn was used to assess the complexity of postural control signals under different conditions."

We hope these revisions address your concerns and improve the clarity and accessibility of our manuscript. Thank you again for your valuable feedback.

9.On line 516 they make comments about “what existing literature indicates” but there are no references supplied. Similarly, on line 521 they comment that their findings “aligns with findings from prior studies” but supply no references.

ANSWER:Thank you for your review, we have already been supplemented with the refs.

10.On line 542, they make the statements that their study “offers actionable insights for practical applications”.  I assume they are attempting to make the statement that their findings are clinically relevant, but again the syntax used is difficult to understand.

ANSWER:Thank you for your review, It has been modified.

11.In the first part of section 4.4, their “comprehensive synthesis of findings” is reasonably well written but far too detailed.  On line 573, they stated their findings are “consistent with central resource theory”.  I am not familiar with this term and there is no discussion of it or a reference for it.

ANSWER:Thank you for your review.The corresponding part has been modified.

12.The “limitations” section is reasonably well written.  However their “contributions to the field” section is far too run on.

ANSWER:Thank you for your review.The corresponding part has been modified.

Thank you again for your valuable feedback. We will make the necessary revisions to our manuscript based on your suggestions, and we believe that these changes will enhance the quality and impact of our work.

Best regards,

Yao Sun, Yingjie Sun, Jia Zhang, and Feng Ran

Reviewer 2 Report (New Reviewer)

Comments and Suggestions for Authors

Dear authors.
Thank you for the opportunity to get acquainted with your interesting research.
Postural maintenance is a very important indicator of health. It should be noted that there is a close indicator of health. This indicator is "static balancing", which is included in many tests for assessing biological age based on health indicators
Recommendation for authors. The authors could reflect this idea in the literature review of their article. We think that this would expand the meaning of the work.
According to the results of the experiment, the authors obtained the effect that with closed eyes the amplitude of oscillations in the subjects is greater than with open eyes. This is also interesting, it could also be compared with the results of studies of static balancing, which is performed with closed eyes. But this recommendation is not mandatory. In my opinion, it would be interesting.

I read the description of the participants in the experiment with great interest. I want to note the very high-quality description of how the selection of young men for the experiment was carried out. The quality of the selection increases the reliability of the results obtained.
According to the description, the experiment was also carried out at a very high level. I can note the high-quality preparation of the young men before participating in the experiment (quality sleep of at least 7 hours during the day, abstinence from alcohol and nicotine, etc.)
Question. Was the fulfillment of these conditions somehow controlled by the young men?
The experimental intervention was also carried out very well. I would like to note that the authors described in detail how exactly the subject performed the cognitive fatigue task (how he sat, at what distance, where he held his hand, etc.). When I read the article, I had such questions, but the authors answered them before I had time to ask.
Description of results. It is well done. But it seems to me that the authors describe unreliable results in too much detail (p>0.05).
Recommendation. Remove all high-quality descriptions of the results (p>0.05). It is only necessary to write that in this case there is no effect of the intervention.
Only the results where p<0.05 need to be described in detail.
The discussion of the results is done at a good level.

Author Response

Dear Reviewer 2, Thank you very much for your insightful comments and suggestions on our manuscript. We greatly appreciate your time and effort in reviewing our work, and we found your feedback to be very constructive.   Regarding the suggestion to include "static balancing" in the literature review: You are absolutely right that "static balancing" is a crucial indicator of health and is often used in assessments of biological age.Regarding your suggestion to include "static balancing" in the literature review, we understand the importance of this indicator in health assessments. However, since our experiment primarily focused on dynamic balance under various sensory interference conditions, we have chosen to mention static balancing in the future work section instead. We believe this will provide a broader context for our findings and highlight potential areas for further research. This will indeed expand the meaning of our work and provide a broader context for our findings.   Regarding the comparison with static balancing studies: We found your suggestion to compare our results with those of static balancing studies performed with closed eyes to be very interesting. Although this was not initially planned, we will consider this comparison in the discussion section of our article, as it could provide additional insights into the effects of mental fatigue on postural control.   Regarding the description of participant selection: We are glad that you found our description of participant selection to be of high quality. We believe that a rigorous selection process is essential for ensuring the reliability of the results, and we are pleased that our efforts in this regard were recognized.   Regarding the control of pre-experiment conditions: We appreciate your question about how we controlled the fulfillment of pre-experiment conditions by the participants. In response to this, we would like to clarify that the experimental staff reminded, recorded, and inquired about the participants' adherence to these conditions daily, achieving a tracking effect. This helped ensure that the participants met the required conditions before the experiment.   Regarding the description of results: We understand your concern about the detailed description of results with p>0.05. You are correct that these results are not statistically significant, and we will revise the manuscript to remove the detailed descriptions of these results. We will only describe the results where p<0.05 in detail, as these are the ones that indicate a significant effect of the intervention.   Regarding the discussion of results: We are pleased that you found the discussion of the results to be well done. We will continue to refine this section to ensure that it provides a clear and comprehensive interpretation of the findings.   Thank you again for your valuable feedback. We will make the necessary revisions to our manuscript based on your suggestions, and we believe that these changes will enhance the quality and impact of our work.   Best regards,   Yao Sun, Yingjie Sun, Jia Zhang, and Feng Ran

Round 2

Reviewer 1 Report (New Reviewer)

Comments and Suggestions for Authors

please see my comments in the word file I have uploaded

This manuscript is a resubmission of an earlier submission. The following is a list of the peer review reports and author responses from that submission.

Round 1

Reviewer 1 Report

Comments and Suggestions for Authors

This paper investigates the effects of a mentally fatiguing stroop task on bipedal balance in young healthy men. The paper confirms existing research that there is no effect of MF on standing balance in young, healthy men.

The paper is very hard to understand in its current state, as there are some serious details missing in the method section. These details makes it very hard to judge the scientific content when information about how the data was processed is missing, the protocol does not provide sufficient information, and the task description is not clear. Additionally, the statistics employed does not take into account the large number of t-tests run and no correction for multiple comparisons was made. Below follow some input in no particular order.

Lines 55-72 and 72-90 are the exact same.

Sometimes word spacing is off or spaces are missing completely – this will probably change with editing.

Participants: non-professional athletes and ‘lacking systematic sports training experience’ are not agreeable. Was this a student population?

Methods: Normally methods are written in past-tense. Generally the authors change between present and past tense in the method section which makes it hard to read.

I don’t understand the description of the Stroop task. You explain that there is a 100% incongruent consistency but this does not match the description where the characters/colours for yellow/green/blue are always congruent but only the red is not. It is further not clear to me how the red was incongruent with this description: “When the character for "red" appeared in red font […]”.

How were the outcome measures determined based on the raw data? No information is provided as to the processing of especially the COP data. Some information on how the COP data was processed (was it filtered, how was velocity determined, etc) is imperative to understand the results.

The experimental flow chart (figure 3) indicates that the post-assessment of balance was twice as long as the pre-assessment of balance. Why is this? I cannot find any explanation in the text.

It would be helpful to the reader if you present the protocol at the beginning of the method section instead of the end. Upon further reading it seems like important measurement moments are missing from the protocol figure and text (there is a post-fatigue measurment at 20 minutes according to the result section that does not appear in the method section). This should be amended.

Statistics: Several T-tests have been performed to compare the COP outcome measures. However, there is no indication that the p-value has been adjusted to reflect multiple comparisons. Please adjust and report with a correction for multiple comparisons.

The order of the outcome measures presented in the result section varies from the order in the method section. Aligning these two will help the reader immensely.

In the result section is stated that 20 minutes post the MF intervention. However, this measurement moment does not appear in figure 3.

Figures: sometimes the ylabel is turned away from the graph and other times towards the graph. Please be consistent. Adding the individual data-points to the barplots will strengthen the visual representations. Colours might also improve readability.

Tables: highlighting the significant p-values help the reader when looking at the table.

Improving the method section and the result section will allow for a proper assessment of the scientific quality of the experiments.

Comments on the Quality of English Language

Be careful with past and present tense as these are used interchangeably. It makes it harder to read.

Reviewer 2 Report

Comments and Suggestions for Authors

General Comments to Authors:

The goal of this paper is to evaluate the impact of cognitive fatigue induced by a 45-minute Stroop task on postural control (i.e., center-of-pressure measures). The authors suggest that cognitive fatigue negatively impacts on postural control measured by sample entropy and frequency-band energy proportions.

Although I acknowledge the contribution of this study, I have a serious concern about the quality of the manuscript to communicate with general readers. I would like to recommend that the authors need to rewrite the manuscript, especially Introduction and Result section (both contents and English). In Introduction, the authors need to verify importance of this study by revealing a void in previous study (or our knowledge) and suggest why this study is important and where this study stands. Unfortunately, the content and length of introduction was limited to fully understand the author’s intention. So, I would like to ask the authors to closely discuss about the previous studies relating to the topic of this study and point out the limitation of the studies.   

In addition, some methodological flaws should be revised or proved by the authors before publication. I believe another weakness of this study is statistical analysis. Since there are many dependent variables, MANOVA should be performed to control family-wise error rate.

Specific Comments to Authors:

Throughout Manuscript

1)     Please unify the terminology. The authors write cognitive fatigue in the Title but write mental fatigue. These terms were also used interchangeably in the manuscript. Although the two terms are used interchangeably, it is considered appropriate to unify the terminology in this paper.

2)     There is too much unnecessary numbering throughout the manuscript.

Title and Abstract

3)     Title: ‘CoP Sensors’  is confusing to me. The authors measured CoP using force platform. Please consider revising this word.

4)     Abstract: Abstract should be a stand-alone, concise document that includes background, aims, methods, results (with summary statistics or p-values), and conclusions. All of these are included, but not concisely communicated. For example, there is too much content written – especially for the results and conclusion in Abstract. Based on my search, Abstract word limit is less than 200 words in this journal.

Introduction

5)     Lines 55-61: Hard to follow; what the intention was of writing this. One of alternative might be to begin with a definition of mental fatigue (MF).

6)     Lines 73-90: The same contents from Line 55-72 is repeated here. I ask all three authors to read the revised manuscript after submitting the paper and give final approval.

7)     Lines 91-107: In the second paragraph of the Introduction, it would be appropriate to discuss previous studies on postural control according to cognitive effects. At present, this topic is slightly beyond the scope of the study, and the general content is presented without references. There are numerous studies related to this topic. Here are some of the list:

Dault, M. C., Frank, J. S., & Allard, F. (2001). Influence of a visuo-spatial, verbal and central

executive working memory task on postural control. Gait & posture, 14(2), 110-116.

https://doi.org/10.1016/S0966-6362(01)00113-8

Kang, S. H., Lee, J., & Jin, S. (2021). Effect of standing desk use on cognitive performance

and physical workload while engaged with high cognitive demand tasks. Applied

Ergonomics, 92, 103306. https://doi.org/10.1016/j.apergo.2020.103306

Kerr, B., Condon, S. M., & McDonald, L. A. (1985). Cognitive spatial processing and the

regulation of posture. Journal of Experimental Psychology: Human Perception and

Performance, 11(5), 617. https://psycnet.apa.org/doi/10.1037/0096-1523.11.5.617

Schwartz, B., Kapellusch, J. M., Schrempf, A., Probst, K., Haller, M., & Baca, A. (2017). Effect

of alternating postures on cognitive performance for healthy people performing sedentary

work. Ergonomics, 61(6), 778-795. https://doi.org/10.1080/00140139.2017.1417642

Stephan, D. N., Hensen, S., Fintor, E., Krampe, R., & Koch, I. (2018). Influences of postural

control on cognitive control in task switching. Frontiers in psychology, 9, 1153.

https://doi.org/10.3389/fpsyg.2018.01153

8)     Lines 112-114: The reasons why the research objective is to be written/addressed using this methodology (i.e., DWT method) should be described based on the results of previous studies. In the current manuscript, there is no rationality to employ these measures. In addition, Independent variables such as eye open/close and proprioceptive perturbation should also be explained based on previous research.

Method

9)     Line 122: Object >>> Participants

10) Lines 133-138: Please provide other parameters (e.g., number of measurements; correlation among repetitive measures; number of groups, etc.) for G power test and provide the reference for this analysis.  

11) Lines 143-151: Unnecessary numbering.

12) Lines 154 (2.2.2. Experimental Setup): Please consider dividing this section into experimental procedure and experimental design.  It is difficult to follow the contents because there are too many numberings in 2.2.2. section.

13) Line 242: It is well known that the stroop test requires executive function. I am wondering if it is possible to measure objective cognitive fatigue (i.e., To be more precise, executive function fatigue) with the Psychomotor Vigilance Task (PVT), which mainly measures attention?  Please provide evidence.

14) While looking at Figure 3, post-fatigue session was about 40min. Then, cognitive fatigue can be recovered as a function of time. How can you address this effect?  Have you checked the order effect?

15) Line 256: Please consider dividing this section into data analysis (or data processing) and statistical analysis.

16) Line 257: The authors should describe in detail how CoP measures were processed to ensure reproducibility.

17) Line 266: Since there are multiple dependent variables, please use MANOVA first and then use subsequent ANOVA.

18) Lines 268-271: This information is unnecessary in this section.

19) Lines 272-274: provide just statistically significant alpha level. It may be sufficient to write the asterisks description in the Note of the Tables.

Results

20) Revise your Results section; each paragraph starts with a clear and concise statement of the main finding, in stead of describing the content in a figure or table. For example, Lines 279-281, Lines 315-317, 326-327, 337 can be changed - please use parentheses. The current Results section simply lists the results, making it difficult to determine which results are the main points in this study.

Discussion

21) Line 427: high-precision  - what does that mean?  Is there any low-precision CoP indicators?

22) Line 429: You mention ‘biomechanical data’ – but I am not sure these measures are real biomechanical measures. For me, these are postural control measures or potentially motor control data.

23) Lines 431-434: No need to list up these variables.

24) Lines 435: Sample entropy first appears in the method section, but shouldn't it have been explained in advance why this measure is essential to this study? (In the Intro)

25) Lines 438: again, this is also needed to be discussed in Introduction first – why this is important in this study?

26) Lines 453: Since increased sample entropy denotes irregular CoP (mostly correlated to decreased variability), please re-express “the variability of sample entropy is high”. This sentence can be confusing.

Conclusion

27) Generally in conclusion, I think the authors have no need to use numbering. Please write the conclusion briefly/concisely – convey main points.  In addition, the contents of section 5.4 should be discussed in the Discussion.

Comments on the Quality of English Language

please see my general comments.

Reviewer 3 Report

Comments and Suggestions for Authors

Introduction to the Review

The article addresses a novel and highly relevant topic, exploring the impact of mental fatigue on postural control. It stands out for its focus on multisensory interaction and the use of advanced technologies such as the force platform and Center of Pressure (COP) signal analysis. Furthermore, the application of the Discrete Wavelet Transform (DWT) to decompose sensory signals and the statistical techniques employed enhance the depth of the work.

Positive Aspects of the Manuscript

  1. Novelty of the Topic:
    The research focuses on a little-explored area, namely the influence of cognitive fatigue on postural control mechanisms, providing relevant data for future studies on multisensory compensation.
  2. Methodological Depth:
    • The use of the force platform and its evaluation methodology to capture precise biomechanical indicators is a strong point of the study.
    • The application of DWT to decompose COP signals offers a detailed analytical level that is useful for understanding underlying processes.
  3. Statistical Analysis:
    The statistical methods employed are appropriate for evaluating variations in data before and after the intervention, demonstrating a high level of statistical rigor.
  4. Experimental Intervention:
    The experimental design is well-structured, with a detailed protocol that ensures the study's replicability. The use of the Stroop task to induce cognitive fatigue is suitable.
  5. Results and Discussion:
    The results are clear and presented in an understandable manner, with a detailed analysis of the metrics obtained. The discussion accurately explains the findings, highlighting how mental fatigue impacts postural control and multisensory compensation mechanisms.

Aspects to Improve

  1. Abstract:
    The objective of the study should be explained more concretely and clearly, specifically highlighting how multisensory interaction and changes in postural control due to cognitive fatigue will be analyzed. A brief explanation of the statistical methods used should also be included. This would help readers immediately understand the significance of the work.
  2. Introduction:
    • Rewrite Paragraph 108-119: This section, which introduces the purpose and methods of the study, could benefit from more fluid and structured writing. Avoid referencing the work that will be read later, as the current structure feels somewhat dense.
    • Explanation of DWT: It would be beneficial to include a brief description of the Discrete Wavelet Transform (DWT) method in the introduction, explaining what it is, how it works, and why it was chosen for this analysis.
  3. Participants:
    The manuscript mentions that the sample size was based on previous research, but it does not cite the specific references supporting the decision to select 26 participants.
    This weakens the justification for the sample size.
  4. Instruments and Equipment:
    • The force platform and E-prime software are critical elements of the study. However, there is no bibliography provided to support their selection or describe their features in detail. Including such references would improve the transparency and rigor of the methodological section.
  5. Comparison with Other Studies:
    Although the results and discussion are clear and detailed, there is a lack of broader comparison with other studies on fatigue, even if they do not use force platforms. Including these references would enrich the context of the work and further highlight its unique contributions.

Final Recommendations

The article has a solid focus and makes significant contributions to the field of postural control and cognitive fatigue. However, the following actions are suggested to strengthen the manuscript:

  1. Revise and clarify the study's objective in the abstract.
  2. Improve the writing of paragraph 108-119 for greater clarity.
  3. Briefly explain the DWT methodology in the introduction.
  4. Include specific references to justify the sample size.
  5. Add bibliographic references for the instruments used, such as the force platform and E-prime software.
  6. Expand the discussion by comparing the findings with previous studies on fatigue, even those not using force platforms.

If these issues are addressed, the manuscript will reach an excellent level, improving its impact and readability for readers.

Reviewer 4 Report

Comments and Suggestions for Authors

-The abstract is very long, not focused on the background, and there is no clear conclusion. 

-The introduction lacks enough depth on the background information. The contents in the first paragraph are repeated. There is no evidence to explain the gaps or rationales of this study and all information looks broad. The choice of selecting healthy young people for postural tasks is not new and there are many publications regarding sensory interference and fatigue on postural skills. This section requires a comprehensive and specific literature review and addresses the key gaps in previous studies.   

-Methods (participants): Please explain if the participants completed a consent form and where the ethics was approved.

-Results: Some sections are very long and there is unnecessary information. The format of charts should be consistent and standard. It seems that the same data is presented in both figures and tables which is redundant. There are many terms in the results that have not been explained in the introduction and also need a clear definition in the methods. Adding a table to define all terms in the appendix would be helpful for readers. 

-Discussion: The first paragraph is the repetition of previous sections or should be moved to the method section. The first paragraph should outline the aim and scope of this study. 

-I am not sure all discussion points can be referred to as mental fatigue because the whole experiment was lengthy and with many tasks that could affect the results as confiding variables. Despite having rest periods, the inter-individual variability might change the motor behaviours. 

-Overall the discussion is really long and descriptive and without any explanation of underlying mechanisms and how different datasets can inform the practice. The manuscript requires a more holistic approach to synthesis the findings and explain the rationales and applications.       

Comments on the Quality of English Language

Please revise the manuscript for clarity.